# Lymph Vessels Associate with Cancer Stem Cells from Initiation to Malignant Stages of Squamous Cell Carcinoma

**DOI:** 10.3390/ijms241713615

**Published:** 2023-09-02

**Authors:** Anna Cazzola, David Calzón Lozano, Dennis Hirsch Menne, Raquel Dávila Pedrera, Jingcheng Liu, Daniel Peña-Jiménez, Silvia Fontenete, Cornelia Halin, Mirna Perez-Moreno

**Affiliations:** 1Section for Cell Biology and Physiology, Department of Biology, University of Copenhagen, 2100 Copenhagen, Denmark; 2Unidad de Investigación Biomédica, Universidad Alfonso X el Sabio (UAX), Avenida de la Universidad 1, Villanueva de la Cañada, 28691 Madrid, Spain; 3Institute of Pharmaceutical Sciences, ETH Zurich, 8093 Zurich, Switzerland; cornelia.halin@pharma.ethz.ch

**Keywords:** cancer stem cells, lymph vessels, skin squamous cell carcinoma, tumor microenvironment

## Abstract

Tumor-associated lymph vessels and lymph node involvement are critical staging criteria in several cancers. In skin squamous cell carcinoma, lymph vessels play a role in cancer development and metastatic spread. However, their relationship with the cancer stem cell niche at early tumor stages remains unclear. To address this gap, we studied the lymph vessel localization at the cancer stem cell niche and observed an association from benign skin lesions to malignant stages of skin squamous cell carcinoma. By co-culturing lymphatic endothelial cells with cancer cell lines representing the initiation and promotion stages, and conducting RNA profiling, we observed a reciprocal induction of cell adhesion, immunity regulation, and vessel remodeling genes, suggesting dynamic interactions between lymphatic and cancer cells. Additionally, imaging analyses of the cultured cells revealed the establishment of heterotypic contacts between cancer cells and lymph endothelial cells, potentially contributing to the observed distribution and maintenance at the cancer stem cell niche, inducing downstream cellular responses. Our data provide evidence for an association of lymph vessels from the early stages of skin squamous cell carcinoma development, opening new avenues for better comprehending their involvement in cancer progression.

## 1. Introduction

Skin squamous cell carcinomas (SCC) are distressingly common and have the ability to metastasize [1]. Whole-genome sequencing studies have provided insights into the SCC mutational landscape [2,3], which is highly conserved between human and mouse skin [4]. Modeling the stages of skin carcinogenesis in mouse skin with chemical carcinogens, such as the widely used two-stage skin chemical carcinogenesis model involving dimethylbenz[a]anthracene (DMBA) and 12-O-tetradecanoylphorbol-13-acetate (TPA), has revealed similar recurrently mutated genes as those observed in human SCC [4]. The skin lesions of the DMBA/TPA carcinogenesis protocol appear as benign papillomas and other benign lesions that progress into squamous cell carcinomas (SCCs) [5,6,7]. This model, in combination with mouse genetics, has exposed the essential roles of driver genes that partake in the initiation, promotion, and progression into malignancy [5,6,7]. These approaches have also provided platforms to identify the cells within the skin, such as hair follicle stem cells (HFSC), that could initiate SCC when carrying driver gene mutations, self-renew, and sustain tumorigenesis—defined as skin cancer stem cells (CSCs) [8,9]. However, the skin mutational burden alone is insufficient to initiate tumorigenesis [10,11], and mounting data indicate that it requires a chronic tumor-supportive microenvironment [12,13,14]. 

In recent years, several advances have provided insight into the composition and contribution of the complex milieu in which skin CSCs reside [15,16,17]. Skin CSCs establish signaling crosstalk with different cell types, including neighboring cancer cells [18], blood vasculature [19,20], and immune cells [21,22]. Clonal CSCs evolve into malignancy in this complex microenvironment, acquiring additional genetic traits that promote heterogeneous cell plasticity states, local invasion, and metastatic dissemination [23,24,25,26,27]. SCC metastatic cells disseminate to tumor-draining-lymph nodes through intratumoral lymph vessels (LVs), spurred by the lymphangiogenic vascular endothelial growth factor C (VEGF-C) [28]. 

Despite the relevance of LVs in skin tumorigenesis, whether LVs interact with the CSC niche remains elusive. Additionally, it is unknown whether such interactions only occur at the late SCC stages. Interestingly, the blockade of the lymphangiogenic factors VEGF-C and VEGF-D inhibits inflammatory skin carcinogenesis [29], supporting the contribution of LVs to tumor development. However, the distribution of LVs at the CSC niche and their potential involvement in tumor initiation is unknown.

In the skin, dermal LVs distribute into two plexuses termed the upper and lower horizontal plexuses [30,31]. Capillary LV networks extend to the upper dermis, whereas collecting LVs reach the lower dermis and the uppermost area of the subcutaneous tissue. We, along with other groups, recently uncovered that LV capillaries are associated with the HFSC niche [32,33,34], regulating HFSC activation state and skin regeneration [32,33].

The fact that SCC can originate from HFSC [8,9] suggests the potential association of LVs and the CSC niche from the initial stages of tumor development. In this study, we investigate whether LVs associate with the CSC niche in benign and SCC skin lesions. We also assess the LV–cancer cell association using cell lines representing the initiation and promotion stages of mouse skin carcinogenesis. Our findings reveal that LV localization at the CSC niche is an early event already present in benign skin lesions. In cell culture, LVs establish direct heterotypic interactions with initiated and promoted cancer cells, without changing their proliferative behavior. Transcriptome analyses exposed cell adhesion, vessel remodeling, and immunity as the top categories involved in their direct crosstalk. These findings reveal that LVs integrate at the CSC niche through adhesive interactions and emphasize the influence of the cancer–LV crosstalk in regulating cancer-cell-driven LV remodeling and immune modulatory cues. Our results pave the way to explore further the contributions of LVs associated with the CSC niche to skin carcinoma initiation, development, and stemness.

## 2. Results

### 2.1. LVs Distribute at the CSC Niche in Mouse and Human Benign and Malignant Skin Lesions

To investigate the relative localization of LVs to CSCs, we conducted histopathological and immunofluorescence microscopy analyses (IFM) on mouse skin sections from benign papillomas and keratoacanthomas, along with SCC (Figure 1A–D) obtained from mice exposed to the two-stage DMBA/TPA skin chemical carcinogenesis protocol. Representative lesions of papillomas displayed a keratotic wart-like organization, whereas keratoacanthomas exhibit a dome shape with a central keratinized crater and squamous down growths with hyperplastic cells [35,36] (Figure 1A,B). SCC exhibited keratinized pearls with well-differentiated areas and regions with elongating edge ridges projecting to the dermis with atypical squamous cells showing hyperchromatic nuclei with mitosis and high nuclear-to-cytoplasmic ratio [37] (Figure 1A,B). 

To evaluate the presence and distribution of LVs within mouse skin lesions, we conducted IFM analyses, utilizing markers for LVs (LYVE-1, lymph vessel endothelial hyaluronan receptor 1) [38], cancer stem cells (CSCs) (SOX2, nuclear SRY-Box Transcription Factor 2) [39,40], and basal progenitor keratinocytes (K5, Keratin 5) [41], which, in the context of skin carcinomas, showcase uncontrolled growth, extending into suprabasal layers. Our findings revealed that LV capillaries were predominantly localized at the tumor–stroma border (Figure 1C) and also expressed the bonafide markers VEGF Receptor 3 (VEGFR3) [42] and the PROX1 transcription factor [43] (Appendix A). LVs were closely juxtaposed to distinct areas of CSCs expressing nuclear SOX2 across all types of skin lesions (Figure 1C and Appendix A). The surrounding perilesional regions, which exhibited a normal epidermal architecture and lacked SOX2 expression, served as internal controls, highlighting the absence of direct LV proximity to the epidermis (Appendix A). LV capillaries showed proximity to areas with hair follicles when present, as previously documented [32,33,34]. In specific regions, LV capillaries extended and infiltrated the tumor–stroma border, establishing contacts with CSCs at these discrete locations (Figure 1D). No significant global changes in the overall density of peritumoral LVs were observed among the different types of skin lesions: papillomas, keratoacanthomas, and SCC (Appendix A). 

To determine whether LVs were predominantly associated with nuclear SOX2^−^ positive CSCs rather than with all cancer cells, we quantified the relative distances between LV with SOX2^+^ cells, as well as between LV with K5^+^ SOX2^−^ cells. We observed that the closest association occurred between LVs and CSCs presented as aggregated quantifications (Figure 1E), since no discernible differences were observed among the various types of skin lesions.

Moreover, to provide functional support for the indication that LVs distributed within the CSC niche, we employed skin sections from K15-CrePR1; Wls^fl/fl^ (Wls^∆K15^) genetically modified mice and control mice subjected to the two-stage DMBA/TPA skin carcinogenesis protocol. The genetic knockout of Wls in hair follicle Keratin 15+ cells results in reduced tumor growth, incidence, and reduced SOX2-positive CSC cell numbers in the skin [44]. Strikingly, the decrease in CSC numbers was also associated with the absence of LV distribution at epidermal sites (Figure 1F).

To evaluate if LVs distributed at the CSC niche in human skin, we analyzed human SCC biopsies from the early T1 to the T2 stage, with either grade 1 or 2 (Appendix A) [45]. Similarly to the observations in mouse skin, LV localized in the vicinity of CSCs more closely than to single labeled K5-positive cancer cells, remaining consistent across the different analyzed stages (Appendix A). These observations indicate that the LVs are part of the CSCs in skin lesions at different stages and grades, suggesting the existence of signaling crosstalk and functional interactions between them, starting already in tumor lesions with lower recurrence and minimal risk of metastasis and death [46,47]. 

### 2.2. LECs and Cancer Cells Do Not Reciprocally Influence Their Proliferation

Seeking to address the direct role of lymphatic endothelial cells (LECs) in the CSC niche, we turned to in vitro studies due to the complexity of intercellular connections in skin tumors. We established co-cultures of LECs and skin carcinoma cell lines to allow indirect and direct intercellular communication (Figure 2A). For this purpose, we used the immortomouse-isolated dermal LEC line [48] and two mouse skin squamous carcinoma cell lines, PDV and HaCa4 [49,50,51,52], both of which express the CSC marker SOX2 (Appendix A). The sole expression of SOX2 is required for maintaining the tumor-initiating properties, being more than a marker but a critical factor for CSC functions [39,40]. The epithelial PDV cell line represents the initiation stage of the two stage-skin chemical carcinogenesis protocol [7] as it was transformed in vitro with dimethylbenz[a]anthracene (DMBA), carries the oncogenic H-Ras G12V mutation, and can initiate tumors [50,51]. The HaCa4 cell line is derived from mouse skin carcinomas, carrying the oncogenic H-Ras G12V mutation and treated with 12-O-tetradecanoylphorbol-13-acetate (TPA), representing the two-stage skin carcinogenesis promotion stage [49,51]. It exhibits an epithelioid morphology and can initiate tumors that progress to malignancy [50,51]. 

We evaluated the effect of co-culturing LECs and cancer cells on cell proliferation by IFM analyses using the phospho Histone H3 cell proliferation marker (Figure 2B,C). Upon quantification, our findings did not reveal significant differences between the proliferation of LECs and cancer cells growing in individual cell cultures to those in co-culture (Figure 2B,C). Interestingly, LECs remained distributed near PDV or HaCa4 cells in co-culture (Figure 2B,C), but not to the immortalized mouse keratinocyte cell line MCA3D, and appear distinctly segregated. This cell line possesses epithelial characteristics, expresses normal Ras proteins, and is incapable of initiating tumorigenesis [51,52,53] (Appendix A). This observation mirrors the in vivo scenario at the CSC niche compared to non-lesion areas (Figure 1 and Appendix A). Our results show no influence of LECs on cancer cell proliferation, and vice versa, at the analyzed time, despite maintaining their close association, suggesting a potential role for LECs in additional aspects of carcinogenesis.

### 2.3. Gene Expression Profiling Reveals a Reciprocal Regulation of Cell Adhesion and Immunity between LECs and Cancer Cells 

We next conducted transcriptome analyses to gain further insight into the mechanisms underlying the connection between LECs and cancer cells at the CSC niche and their potential roles in tumorigenesis. For these studies, we used the HaCa4 cell line since it initiates a high number of tumors and progresses further into malignancy, thus, comprising a broader spectrum of tumorigenesis features [50,51]. We compared the transcript levels of FACS-isolated HaCa4 cells and LECs from co-cultures with those from individual cell cultures (Appendix A). The differential expression analyses revealed a total of 217 transcripts deregulated in HaCa4 co-cultured with LECs compared to HaCa4 control (142 upregulated, 75 downregulated) (Appendix A). Gene ontology (GO) enrichment analyses revealed that the top category in HaCa4 cells co-cultured with LECs was cell adhesion, followed by angiogenesis and categories related to immune response (Figure 3A). Within the adhesion GO category, genes related to cell adhesion molecules (CAM) and cell-matrix adhesion were found to be deregulated. Notably, these adhesion genes are connected with angiogenesis and immune response modulation, suggesting a potential integration of broad responses. Upregulated genes included the cell adhesion molecules *Icam1*, *Itga4*, and *Selp*, whereas *Itga2* was downregulated [54] (Figure 3B). Additionally, the transcriptome showed increased expression of the extracellular matrix proteins *Wisp1*, *Cyr61*, *Lgal3bp*, *Nid1*, *Lama4*, *Col5a3*, and *Tinagl1* [55,56,57,58]. Interestingly, *Wisp*, *Cyr61*, and *Lgal3bp* have been described as playing roles in lymphangiogenesis or angiogenesis [59,60,61,62,63], the second top GO category. Among the upregulated genes in the angiogenesis group, with a role in endothelial cell remodeling, we found the VEGF receptor genes *Kdr* and *Flt4* (VEGFR2/VEGFR3) [64] and the secreted factors *Angpt2*, *Adm*, and *Mmp2* [65,66,67,68] along with the chemokines *Cxcl3* and *Cxcl12* [69], which were also present in the third top GO category, immune response. Overall, these findings suggest that LECs drive a global response in cancer cells which, in turn, could further modulate their tumor environment. 

The transcriptome analyses of the LEC, when co-cultured with cancer cells, revealed a reciprocal modulation, with 191 genes deregulated, including 127 upregulated and 64 downregulated compared to controls (Appendix A). GO enrichment analyses showed the prevalence of immune, inflammatory response, and chemotaxis categories (Figure 3C). In the GO category, “regulation of cell shape,” adhesion receptor genes with roles in immune responses and endothelial cell remodeling were significantly upregulated. These genes comprised *Vcam1*, *Pdpn*, *Itga7*, *Sele*, and *Selp* [70,71,72] (Figure 3D). Interestingly, *Vcam1*, *Sele*, and *Pdpn* could potentially associate with genes found in the cancer cell transcriptome, such as *Itga4* and *Lgal3bp* [73,74,75]. Additionally, we identified other upregulated genes involved in angiogenesis, lymphangiogenesis, and LV remodeling, such as the vascular regulators *Hif1a* and *Adm* [66,67,76]. Moreover, we identified genes modulating angiogenesis and immunity, including *Il6*, *Cxcl1*, *Cxcl2*, *Cxcl3*, *Cxcl4*, and *Cxcl5* [77,78,79] (Figure 3D), and genes involved in the broader regulation of soluble and immune cellular responses, such as the *Ccl2*, *Ccl7*, *Ccl9*, *Ccl20*, *Csf1*, *Csf2*, *Csf3*, *Il23a*, and *Tnfsf15*, receptor *Il1r1*, and the transcription factors *Nfkb1*, *Nfkb2*, *Nfkbia*, and *Nfkbiz* [31,80]. These gene changes suggest that cancer cells influence the expression of potential adhesion links in LECs, thereby maintaining their location at the CSC niche. This influence, in turn, may promote endothelial remodeling and the expression of immune cues, potentially sustaining chronic immune responses.

Without excluding the possible relevance of paracrine cues in the reciprocal regulation of gene expression in cancer cells and LECs, we focused on the permanence of LECs in the CSC niche. Therefore, we focused on validating the mRNA expression of genes enriched in the cell adhesion category using RT-qPCR. For these experiments, we analyzed both initiated and promoted cancer cells co-cultured with LECs, to characterize a potential shared signature, along with LECs exposed to cancer cells (Figure 3E–H). The tested adhesion-related genes showed elevated expression in both PDV and HaCa4 (Figure 3E,F). Although the overexpression of genes, including *Lama4*, *Icam1*, and *Itg4*, differed in extent and exhibited a higher overexpression in HaCa4 than in PDV cells, these results suggest that LECs lead to the expression of cell adhesion molecules in cancer cells at both the initial and later stages of skin carcinogenesis. In turn, both initiated and promoted cancer cells induced gene expression changes in LECs, as seen by the overexpression of the analyzed adhesion genes in LECs upon co-culture with the two cell types (Figure 3G,H), although not all yet statistically significant. Furthermore, qRT-PCR analyses of some enriched genes in the angiogenesis and immune response categories were evaluated in both PDV and HaCa4, and a comparable signature was observed in this case as well (Figure 3E,F).

Overall, these results indicate that LECs and cancer cells at different stages of carcinogenesis have a direct mutual influence on the expression of adhesion molecules, potentially sustaining their close association in the niche. 

Moreover, our findings also revealed the effect of their direct crosstalk in fostering the expression of genes involved in vascular regulation and immunity-related responses. 

### 2.4. LECs Establish Heterotypic Cell–Cell Adhesion with Cancer Cells Representing the Initiation and Promotion Stages of Carcinogenesis

The adhesion of skin cancer cells to endothelial cells is a well-established phenomenon that enables tumor cell extravasation in the late stages of carcinogenesis. However, the mechanisms of their bonding during the early stages of cancer development remain unclear. In light of our transcriptome findings, we conducted experiments to assess the functional adherence of LECs to both PDV and HaCa4 cells to ensure that the observed close distribution resulted from direct binding and not random cell crowding. To this end, we established confluent monolayers of LECs and seeded either PDV or HaCa4 cells on top (Figure 4A), allowing them to attach at different time points. After the designated time, we washed out non-adhered cells to gain valuable insights into the speed and efficiency of cell attachment. The results show that LECs can adhere to both PDV and HaCa4 cells (Figure 4B). Interestingly, the kinetics of cell adhesion differed based on the number of adherent cells over time. The PDV cell line exhibited a delayed initial attachment to LECs, compared to the HaCa4 cell line (Figure 4C), potentially attributed to the higher expression of cell adhesion molecules in HaCa4 (Figure 3E,F). However, at later time points, no significant differences were observed, suggesting that cancer cells may exhibit varying levels of expression of the cell adhesion proteins that contribute to their initial association with LECs. These findings reveal the existence of heterotypic interactions between LECs and skin cancer cells at both the initiated and promoted stages of skin carcinogenesis. 

### 2.5. LEC–Cancer Cell Adhesion Induces Reciprocal Changes in Actin Reorganization

Cell adhesion in carcinogenesis is positively and negatively regulated, enabling cancer cells to attach to other cells and the extracellular matrix, favoring their permanence at the CSC niche or their migration and dissemination to distant sites [54]. The actin cytoskeleton plays a critical role in cell adhesion and migration. Therefore, we next sought to identify the occurrence of changes in actin organization at sites of heterotypic cell–cell interaction between LEC and the cancer cell lines. 

In individual cell cultures, LEC cells grew as compact adhesive cell monolayers, displaying a well-defined cortical actin network forming a honeycomb pattern with radial bundles of actin fibers at cell–cell contact sites (Figure 5A,B). On the other hand, epithelial PDV and the epithelioid HaCa4 cells established cell–cell contacts with a polarized actin cytoskeleton organization (Figure 5A,B).

Upon co-culture, we observed distinct morphological changes in both LECs and cancer cells. LEC cells exhibited a more elongated organization and no longer formed a close monolayer. Instead, they organized into an open network, establishing contact with cancer cells. The actin cytoskeleton of LEC and cancer cells appeared reorganized, and areas of directed actin membrane protrusions were evident at heterotypic cell adhesion sites, along with the presence of actin bundles and lamellipodia membrane protrusions (Figure 5A,B). These results underscore the existence of productive adhesive interactions involving the actin cytoskeleton between LECs and cancer cells at both the initiated and promoted stages of skin carcinogenesis.

### 2.6. LECs and Cancer Cells Establish Dynamic but Persistent Heterotypic Interactions

In light of our previous findings, we sought to investigate the dynamics of the heterotypic interactions between LECs and cancer cells more deeply. We conducted time-lapse imaging microscopy of LEC with PDV or HaCa4 cells in co-culture to monitor their adhesive interactions over time (Figure 6A, Appendix A). The movie frames (Appendix A) in Figure 6A,B show the close and persistent association between LEC and cancer cells, as observed in the movie kymographs (Appendix A). Cancer cells located on the top or bottom of LECs exhibited dynamic ruffles and lamellipodia (Appendix A) and occurred in nests or clusters. Some cancer cells sporadically distributed to other sites, while others maintained their dynamic association at these niches, establishing productive cell–cell connections over extended periods. LECs and cancer cells interact with one or more cells over time (Appendix A). We observed that some cells divided during the recorded time, but their quantification revealed no changes in proliferation (Appendix A), in agreement with our prior results (Figure 2). Interestingly, the quantitative assessment of the cells establishing heterotypic adhesion showed a correlation between the surface length of intercellular contacts and their permanence over time (Figure 6C,D). Our results disclose that dynamic cell contacts and actin remodeling are essential for establishing heterotypic adhesions and maintaining LEC association with skin cancer stem cells.

## 3. Discussion

Recent findings uncovered that, in homeostasis, LVs associate in a polarized manner with the hair follicle stem cell niche [32,33,34], regulating stem cell activation states and regeneration [32,33]. In SCC, lymphangiogenic factors play a functional role in driving tumor development and malignancy [28,29]. However, the LV distribution at the CSC niche at distinct tumor stages remains elusive. 

In this study, we show that LVs and CSCs coexist at the mouse and human skin CSC niche from benign and early cancer lesions to late-stage tumors, with LVs distributing at the tumor–stroma borders where CSCs reside (Figure 7). While the presence of SOX2+ CSCs within skin lesions does not exclusively indicate their origin from hair follicle stem cells (HFSCs), our results offer novel insights into the connections between CSCs and LVs, irrespective of the potential cell of origin of CSCs. CSCs exhibit distinct characteristics associated with the tumor microenvironment at these anatomic locations, as previously documented for human [16] and mouse SCC [19,20]. Intriguingly, not all SOX2-positive CSCs interact with LVs, suggesting the existence of specific subsets of CSCs with the capability to associate with LVs. Heterogenous populations of CSCs have, indeed, been defined in SCC at the tumor–stroma borders [16,20]. While functional connections between CSCs and blood vasculature are well documented [19,20,40], less is known about the interactions between CSCs and LVs, and the prognostic significance of LVs in malignant SCC is still controversial [81,82,83].

In our co-culture experiments using mouse cancer cells representing the initiation and promotion stages of skin carcinogenesis, we identified a reciprocal induction of cell adhesion cues involved in LV remodeling and immunity regulation (Figure 7). We found the expression of direct interaction partners in cancer cells and LEC, such as *Itga4* and *Vcam1* [84], which could modulate heterotypic adhesion and maintenance at not only the same niche but also vessel remodeling and immune recruitment [72]. Indeed, *Vcam1* overexpression in LECs and cell binding to the endothelium through VCAM1 have been shown to promote the dynamic opening of LEC junctions [85,86], in agreement with the changes we observed in the organization of the LEC monolayers and actin rearrangements upon co-culture. The observed direct establishment of dynamic heterotypic cell contacts between cancer cells and LECs involving cytoskeletal rearrangements could foster, in skin tumors, fluid and macromolecules’ transport and immunomodulatory responses at the niche [31], together with the expression of secreted factors governing LV permeability, remodeling, and maturation. Consequently, a comprehensive investigation into the intricate interplay of VCAM1 and ITGA4 gains significance due to its potential therapeutic implications across multifaceted functional roles. Their involvement in lymphatic vessel remodeling, shaping immune responses, and steering tumor progression underscores their pivotal function within the tumor microenvironment [87]. Delving deeper into the underlying molecular mechanisms, through techniques like loss and gain of function studies or antibody blockade, offers the prospect of unveiling novel therapeutic pathways that disrupt this interaction. This, in turn, holds potential in addressing CSC maintenance, immune evasion, and the modulation of the microenvironment.

Interestingly, we did not observe changes in cancer cell and LEC proliferation, suggesting that the LV–cancer cell crosstalk, in regulating this aspect of cell behavior, is indirect, possibly favored by other cell types or cues present at the niche. Indeed, our data show an increase in the expression of genes involved in immune response and chemotaxis (e.g., *Il6*, *Cxcl,* and *Ccl* genes), as also detected in LECs stimulated with tumor-derived conditioned medium [88]. In vivo findings also support this notion. The expression of soluble VEGFR-3 blocking the lymphangiogenesis factors VEGF-C/VEGF-D inhibits inflammatory cell recruitment to the tumor microenvironment, reducing skin tumor incidence and growth [29]. Although the direct involvement of LV-derived VEGF-C/VEGF-D has not yet been elucidated, these findings underscored their involvement at the early stages of tumor growth beyond their well-established roles in metastatic progression [28]. The VEGF-C/VEGF-D mediated immune cell recruitment could, in turn, foster tumor initiation by modulating lymphangiogenesis and cancer stemness. In this regard, tumor-associated macrophages secreting VEGF-C/VEGF-D regulate LV density [89] and, in the skin, distinct macrophage populations distribute at the skin CSC niche [21]. Other cell types with well-established roles in skin carcinogenesis, not yet associated with the skin CSC niche, may also be guided to the supportive tumor microenvironment fostered by the CSC–LV connections. While the significance of a CSC–blood vascular niche for tumor growth and stemness is well-recognized [19], LVs could also play a crucial role due to their contributions to fluid and macromolecule transport, immune cell trafficking, antigen presentation, and intercellular signaling. These interconnected processes collectively shape the tumor microenvironment and could influence tumor stemness. Both lymphatic and vascular vessels are integral components of the complex tumor ecosystem, and future research will provide a better understanding of their interconnected roles, contributing to the complex landscape of cancer initiation, CSC maintenance, and cancer development. 

Taken together, our results demonstrate the existence of direct crosstalk between CSCs and LVs that reciprocally influences gene expression programs related to immunomodulatory responses independently of other cells in the complex tumor milieu. Future studies should also focus on identifying specific subsets of CSCs that associate with LVs and on elucidating the roles of molecules involved in the heterotypic CSC–LV interactions and their in vivo function at the CSC niche. Understanding the contributions of these signals to tumor initiation and stemness may offer insights into potential therapeutic strategies to prevent and eradicate cancerous skin lesions from their initial stages, representing a relevant prevention and therapeutic window before SCC progresses to malignancy.

## 4. Materials and Methods

### 4.1. Human and Mouse SCC Samples

Human tissues were purchased from US Biomax now TissueArray (TMA SK-801c, TissueArray, Derwood, MD, USA) or previously donated by Robert Loewe from the Department of Dermatology, Medical University of Vienna [44]. The analyzed mouse samples were derived from mice described in a prior study [44]. Briefly, K15-CrePR1 [90] mice backcrossed with Wls^fl/fl^ mice [91] were subjected to a chemically induced skin tumorigenesis protocol involving the topical application of 7,12-dimethylbenz[a]anthracene (DMBA) followed by phorbol 12-myristate 13-acetate (TPA) to initiate and promote skin tumor development. Tumor development was monitored over 25–30 weeks, and skin lesions were assessed for tumor incidence, size, and histopathological features.

### 4.2. Cell Lines

The immortalized mouse keratinocyte cell line MCA3D [51,52,53] and the mouse skin squamous cell carcinoma cell lines HaCa4 [49,51] and PDV [92] were generously provided by Dr. Amparo Cano (Instituto de Investigaciones Biomédicas, Madrid, Spain). These cells were grown in a medium composed of DMEM and F12 (1:1 ratio), supplemented with 10% FBS (F7524, Sigma–Aldrich, Burlington, MA, USA), and 1% Penicillin/Streptomycin (Gibco, Waltham, MA, USA) and maintained at 37 °C in a humidified environment with 5% CO_2_. The cancer cell lines, even when cultured in a 10% FBS-containing medium, or in a normal calcium concentration, retain their CSC characteristics, being capable of initiating tumors that progress and maintain tumorigenicity upon transplantation into mice [50,51,53].

Lymphatic endothelial cells (LECs), originally derived from the immortomouse strain, carry the Immortomouse transgene, derived from the H-2Kb-tsA58 transgenic mice. This transgene allowed the direct derivation of conditionally immortal LECs and enables inducible expression of a thermolabile large T antigen of the SV40 virus [48]. LECs were cultured in growing permissive conditions, as detailed in a previous study [48]. Briefly, the basal medium was supplemented with 0.5 μL of IFN-γ (PeproTech, Cranbury, NJ, USA) per 50 mL of medium, at 33 °C in a humidified environment with 5% CO_2_, inducing the expression of the SV40 large T antigen. The basal medium consisted of 40% DMEM (supplemented with D-Glucose and pyruvate, Gibco, Waltham, MA, USA), 40% Ham’s Nutrient F12 Medium (with L-glutamine, Gibco, Waltham, MA, USA, and 20% fetal bovine serum (FBS, Sigma–Aldrich, Burlington, MA, USA. The medium also included 1% Penicillin-Streptomycin (100 μg/mL, PS, Gibco, Waltham, MA, USA), 0.2% heparin (Sigma–Aldrich, Burlington, MA, USA), and 0.2% VEGF (US Biological, Salem, MA, USA). For functional assays, LECs were subjected to non-permissive conditions, cultured at 37 °C for 48–96 h without IFNγ. This approach facilitated the degradation of the SV40 large T antigen, thus enabling the cells to adopt a functional phenotype. Two different LEC clones were cultured and no discernible differences in the characteristics or behavior were observed and, thus, they were used equivalently during the study.

Regarding lentiviral infection, HaCa4 and LEC were transduced according to the manufacturer’s instructions with recombinant lentivirus harboring LifeAct^®^-TagRFP/GFP transgene (rLVubi-LifeAct-TagRFP/GFP, ibidi, Gräfelfing, Germany) with 8 µg/mL polybrene (Sigma–Aldrich, Burlington, MA, USA. Stable clones were generated after selection with 2 µg/mL puromycin (Sigma–Aldrich, Burlington, MA, USA).

### 4.3. Immunofluorescence and Antibodies

Paraffin-embedded sections were deparaffinized, rehydrated, treated for antigen retrieval, and stained, as previously described [33]. Hematoxylin and eosin staining was performed following standard protocols.

Optimal cutting temperature (OCT) compound embedded sections were fixed in 4% PFA for 10 min and the antibody stainings were performed as for paraffin-embedded sections. 

Cells plated on coverslips were fixed in 4% PFA for 10 min, permeabilized with 0.3% Triton X-100 PBS for 12 min, and blocked in 2% BSA for 1 h. Primary and secondary antibodies were incubated for 1 h or overnight and 1 h, respectively, at 4 °C in the blocking solution. Nuclei were stained with Hoechst. 

The following primary antibodies were used: anti-Keratin-5 (Polyclonal chicken Poly9059, 1:500, Covance, Princeton, NJ, USA), anti-SOX2 (Monoclonal rat, Btjce 1:100, eBioscience, San Diego, CA, USA), anti-LYVE1 (Polyclonal Goat, 1:100, AF2125 and AF2089, R&D, Minneapolis, MN, USA), anti-PROX1 (Rabbit 11-002P, 1:100, Angiobio, San Diego, CA, USA), anti-VEGFR3 (Goat AF743, 1:100, R&D, Minneapolis, MN, USA), anti-PODOPLANIN (rat, PA2.26, kindly gifted by Dr. Miguel Quintanilla, Instituto de Investigaciones Biomédicas, Madrid, Spain), and anti-phospho Histone H3 (Rabbit 06-570, 1:250, Sigma–Aldrich, Burlington, MA, USA). The following secondary antibodies were used: anti-rat, anti-chicken, anti-goat, and anti-rabbit conjugated to AlexaFluor488 (1:200, Molecular Probes, Eugene, OR, USA), to AlexaFluor 546 or to AlexaFluor 647 (1:200, Jackson ImmunoResearch, West Grove, PA, USA). 

Images were captured using a Nikon Eclipse Ni-E microscope (Nikon Europe, Amstelveen, The Netherlands) adapted with a Photometrics BSI Express camera and CFP plan objectives with the NIS-elements acquisition software (Nikon Instruments, version 5.30.06).

### 4.4. Image Analyses

The acquired images were imported into the ImageJ software (National Institutes of Health, Bethesda, MD, USA). Fixed-size frames and regions of interest (ROI) were used for each image. This standardization ensured consistent analysis of the same area across all samples, facilitating reliable comparisons.

Thresholding: A uniform thresholding approach was employed to segment the ROIs containing lymphatic vessels (LVs), cancer stem cells (CSCs) expressing SOX2, and basal keratinocytes (K5). The threshold level was determined based on the distinctive characteristics of each marker and remained constant for all analyzed images. 

The peritumoral lymphatic vessel density quantification was performed using the Weidner technique [93], as described in prior studies [94,95]. It is based on the determination of the “hot spot” or region of interest (ROI), being defined as the area with the highest density of stained structures when analyzed at a low magnification of 100× (10× objective lens and 10× ocular lens).

### 4.5. FACS Sorting

For the sorting of co-cultured HaCa4 and LEC, and PDV and LECs, cells were stained as previously described [96] using the primary antibody CD31 (Rat MEC 13.3,1:100, B.D. Franklin Lakes, NJ, USA) and the secondary antibody anti-rat AlexaFluor 647 (1:200, Jackson ImmunoResearch, West Grove, PA, USA). HaCa4 or PDV cells were stained with CFSE Cell Proliferation Kit (ThermoFisher Scientific, Waltham, MA, USA) following the manufacturer’s instructions, and CD31^+^ LECs were sorted with an Aria II FACS sorter (B.D. Franklin Lakes, NJ, USA) excluding dead cells and doublets. 

### 4.6. Bulk RNA-Seq Analysis

Samples were sequenced with the Beijing Genomics Institute sequencing (BGISEQ) platform. Raw reads were filtered with SOAPnuke (v1.5.2) software [97]. Filtered clean reads were aligned to the mouse reference genome (GCF_000001635.26_GRCm38.p6) using HISAT (v2.0.4) [98]. Differential gene expression analysis was assessed using DeSeq2 [99]. Genes with log2 fold change ≥ 0.5 and Qvalue ≤ 0.05 were considered differentially expressed. The expression values of each gene were quantified as read counts after normalization. The enrichment analyses, including GO Term analysis, were performed with the web-based RNA Data Visualization and Analysis System, Dr. Tom. The expressed genes related to the categories of interest were plotted as heatmaps with transcript per million (TPM) z-score normalized expression values.

### 4.7. RNA Isolation and qRT-PCR

Total RNA was isolated using the RNeasy Kit (Qiagen, Venlo, Limburg, The Netherlands) according to the manufacturer’s instructions. An amount of 1 µg RNA was used for cDNA synthesis using the Maxima First Strand cDNA synthesis KIT (ThermoFisher Scientific, Waltham, MA, USA). Quantitative PCR assays were performed using 5 ng of cDNA as a template and the reactions were conducted using the GoTaq qPCR Master Mix (Promega, Madison, WI, USA). qRT-PCR was performed on QuantStudio™ 7 Flex Real-Time PCR (Applied Biosystems, Waltham, MA, USA). HPRT housekeeping gene was used for normalization. The gene-specific primer sets were used at a final concentration of 10 µM (Appendix A). All qRT-PCR assays were performed in triplicate in at least three independent experiments. 

### 4.8. Adhesion Assay

LifeAct-RFP-mLECs were seeded and cultured to form a confluent monolayer. PDV cells were stained with CellTrace™ CFSE Cell Proliferation Kit (ThermoFisher Scientific, Waltham, MA, USA) following the manufacturer’s instructions. Either PDV-CFSE or HaCa4-LifeAct-GFP were seeded on top of the monolayer and washed twice with PBS after 15, 30, 45, and 60 min, analyzing biological replicates for statistical analyses. 

### 4.9. Live-Imaging Microscopy

LifeA-ctGFP-HaCa4 and LifeAct-RFP-mLECs were co-cultured or single-cultured in 8-well chamber slides (ibidi, Gräfelfing, Germany) with 10 µM Hepes. Time-lapse experiments were performed with a Nikon Eclipse Ti2 microscope and the NIS-elements acquisition software (Nikon Instruments, Nikon Europe, Amstelveen, The Netherlands) at 20× magnification, with controlled temperature and CO_2_ levels. Images were captured every 3 min over 13 h.

## 5. Statistics

Statistical analyses were performed using Prism 9 (Graphpad) software. n refers to number of samples or biological replicates. Data represent mean  ±  SEM. Exact n numbers for each dataset are detailed in the Figure legends. The normality of the data was evaluated and the statistical significance was calculated by the two-tailed Mann–Whitney test, considering *p*  <  0.05 as statistically significant.

## Figures and Tables

**Figure 1 ijms-24-13615-f001:**
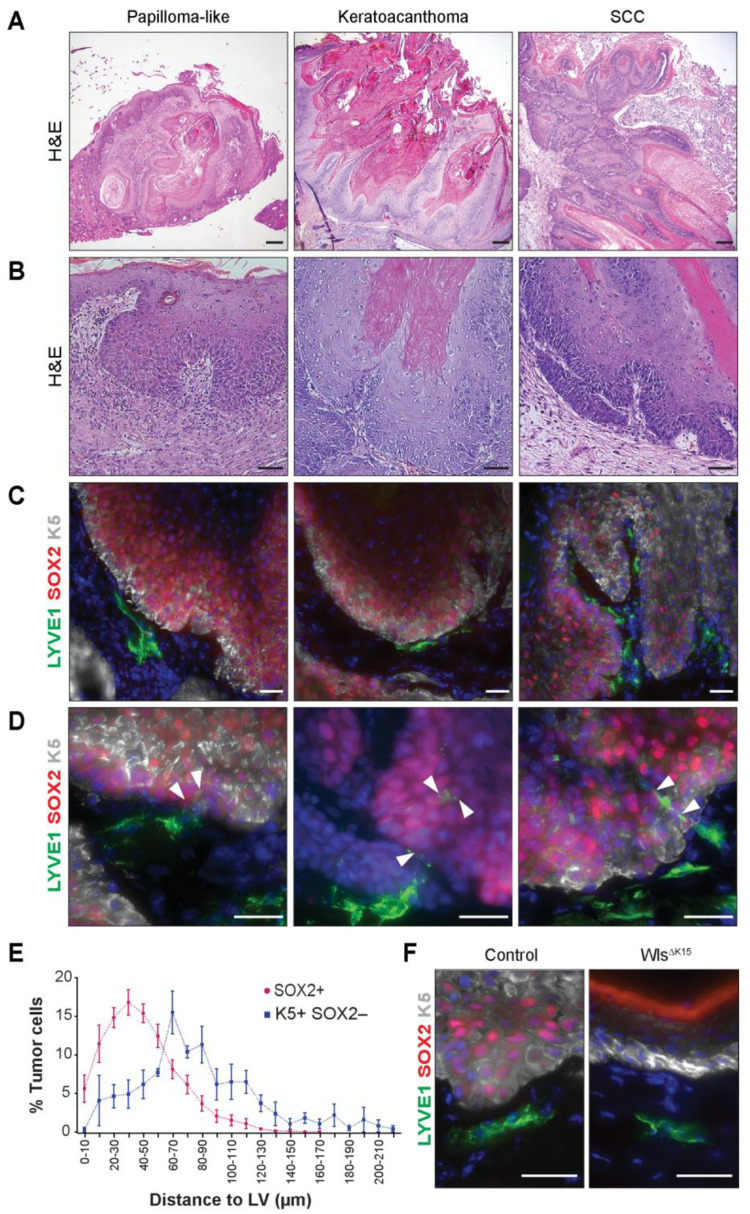
LVs distribute at the CSC niche in mouse benign and malignant skin lesions and infiltrate the tumor–stroma border. (**A**,**B**) Hematoxylin– and eosin–stained sections of mouse skin papilloma, keratoacanthoma, and squamous cell carcinoma tissues. Scale bars = 200 and 50 µm, respectively. (**C**,**D**) Mouse skin immunostaining for the LV marker LYVE1 (green), the CSC marker SOX2 (red), and the basal progenitor keratinocyte marker K5 (white). Hoechst nuclear staining is shown in blue. Arrowheads indicate LV penetrating the tumor–stroma border and contacting CSC at distinct regions. Scale bar = 30 µm. (**E**) Distribution of SOX2+ CSC relative to the distance to LV, n = 10 (3 SCCs, 5 papillomas, 2 keratoacanthoma– like), and distribution of K5^+^ SOX2^−^ tumor cells relative to the distance to LV, n = 4 (1 SCC, 3 papillomas). (**F**) Mouse skin SCC and mouse K15CrePR1^+/T^ Wls^lox/lox^ (Wls^ΔK15^) skin immunostaining for the LV marker LYVE1 (green), the CSC marker SOX2 (red), and K5 (white). Hoechst nuclear staining is shown in blue. Scale bar = 30 µm.

**Figure 2 ijms-24-13615-f002:**
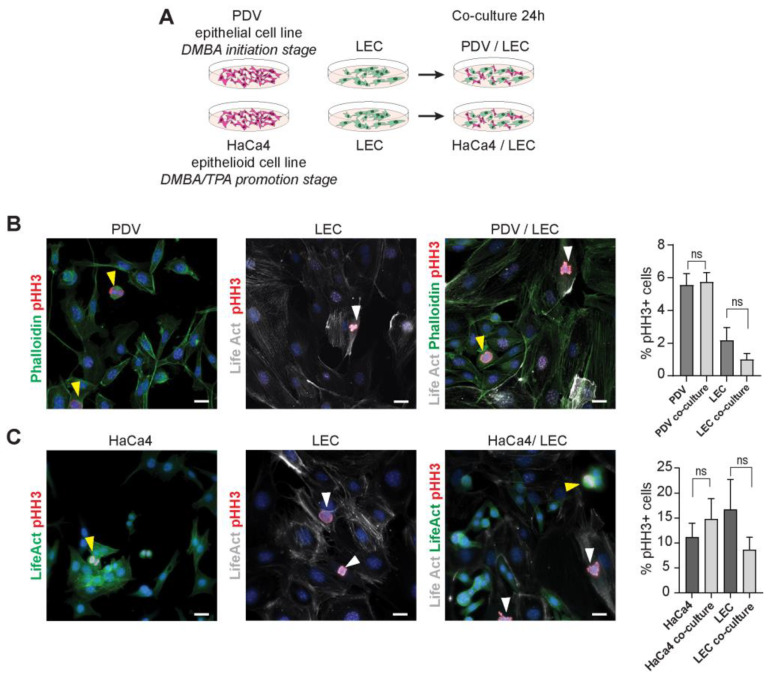
LECs and cancer cells representing the initiation and promotion stages of carcinogenesis do not reciprocally influence their proliferation. (**A**) Schematic illustrating the cancer cell lines PDV and HaCa4, and LECs used in the study with the experimental strategy. (**B**) Immunofluorescence images of single-cultured and co-cultured PDV (phalloidin, green) and LECs (phalloidin, green and LifeAct probe, white) stained for the proliferation markers pHH3 (red) with respective quantifications (n = 6). ns. Not significant. (**C**) Immunofluorescence images of single-cultured and co-cultured HaCa4 (LifeAct-GFP) and LEC (LifeAct probe, white) stained for the proliferation marker pHH3 (red) with respective quantifications (n = 3). Hoechst nuclear staining is shown in blue. White and yellow arrowheads indicate pHH3-positive LECs and cancer cells, respectively. Scale bar = 30 µm.

**Figure 3 ijms-24-13615-f003:**
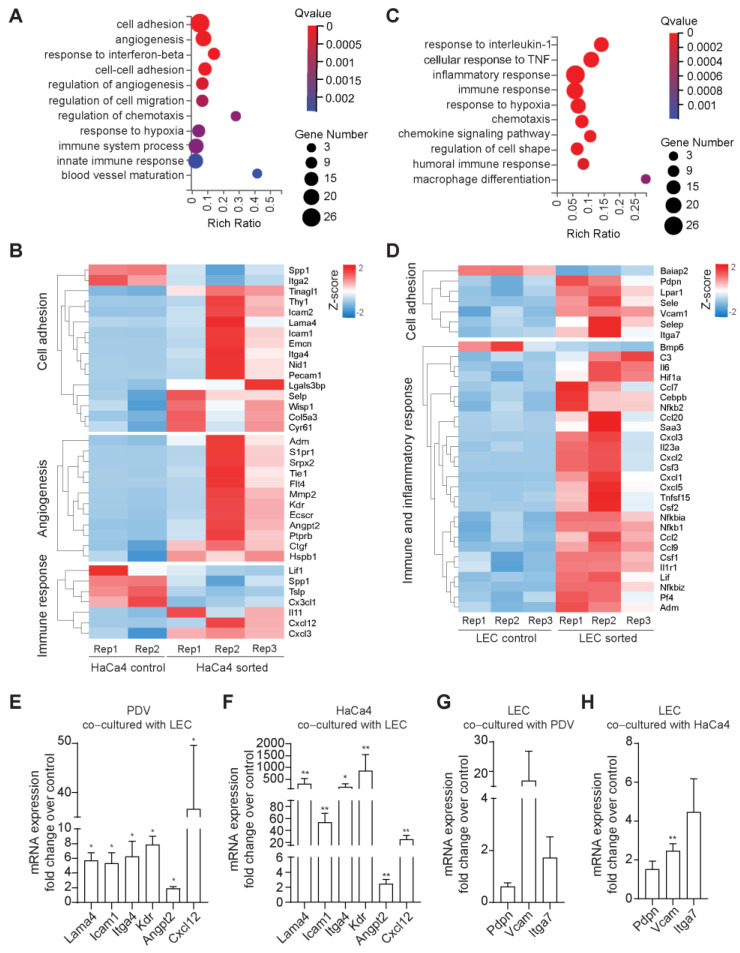
Gene expression profiling reveals a reciprocal regulation of cell adhesion and immunity between LECs and cancer cells. Control cells cultivated as individual cultures, and HaCa4/LEC cells grown in co–cultures were isolated using FACS sorting (HaCa4–sorted and LEC–sorted populations, respectively) (see Appendix A). (**A**) GO pathway enrichment analysis plots of HaCa4–sorted vs. HaCa4 control. (**B**) Heatmaps of differentially expressed genes in HaCa4–sorted vs. HaCa4 control belonging to the adhesion, angiogenesis, and immune-related categories. (**C**) GO pathway enrichment analysis of LEC–sorted vs. LEC control. (**D**) Heatmaps of differentially expressed adhesion– and immune–related genes in LEC–sorted vs. LEC control. (**E**,**F**) qRT– PCR analysis of mRNA expression of genes of the adhesion, angiogenesis, and immunity signatures in PDV–sorted vs. PDV cells and HaCa4–sorted vs. HaCa4 control, respectively. (**G**,**H**) qRT–PCR analysis of mRNA expression of genes of the adhesion signature in LEC–sorted after co–culture with PDV and HaCa4, respectively. n ≥ 3; *, *p* < 0.05; **, *p* < 0.01.

**Figure 4 ijms-24-13615-f004:**
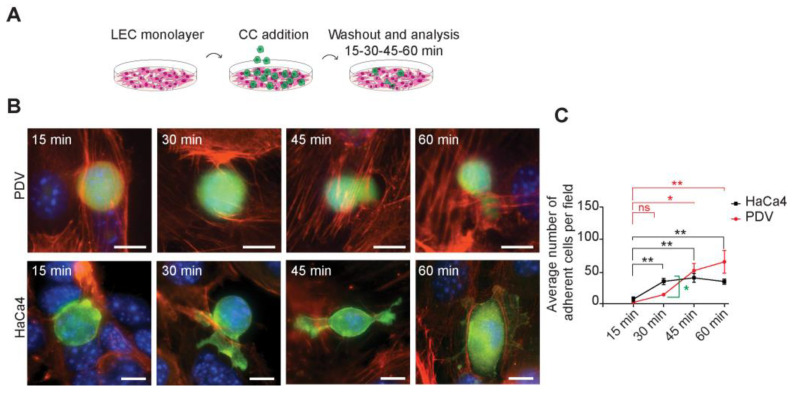
LECs and cancer cells establish heterotypic cell–cell adhesion. (**A**) Schematic illustrating the adhesion assay experimental strategy. (**B**) Representative images of initiated PDV (CFSE, green) and promoted HaCa4 cells (LifeAct–GFP) adhering to LECs (LifeAct–RFP) over time (15–60 min), respectively. Scale bar = 10 µm. (**C**) Quantification of initiated PDV and promoted HaCa4 cells adhering to LECs over time (15–60 min) and comparison of their adhesion kinetics (n = 5). *, *p* < 0.05; **, *p* < 0.01. ns. Not significant.

**Figure 5 ijms-24-13615-f005:**
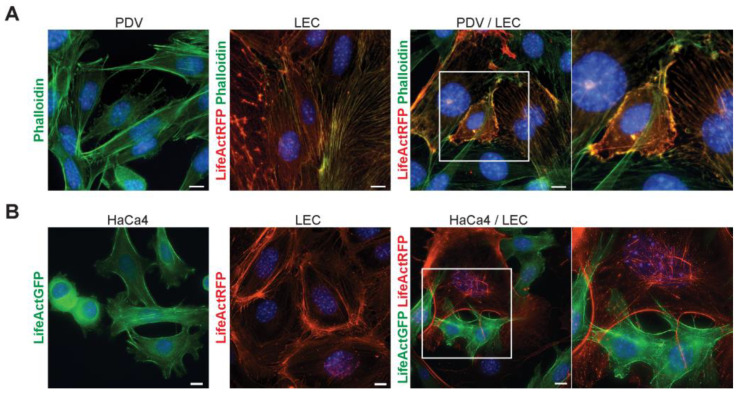
LECs and cancer cells exhibit morphology changes and actin reorganization when co–cultured. (**A**) Representative images of initiated PDV (phalloidin, green) and (**B**) promoted HaCa4 cells (LifeAct–GFP) in single culture and co–culture with LEC (LifeAct–RFP), respectively. Scale bar = 10 µm.

**Figure 6 ijms-24-13615-f006:**
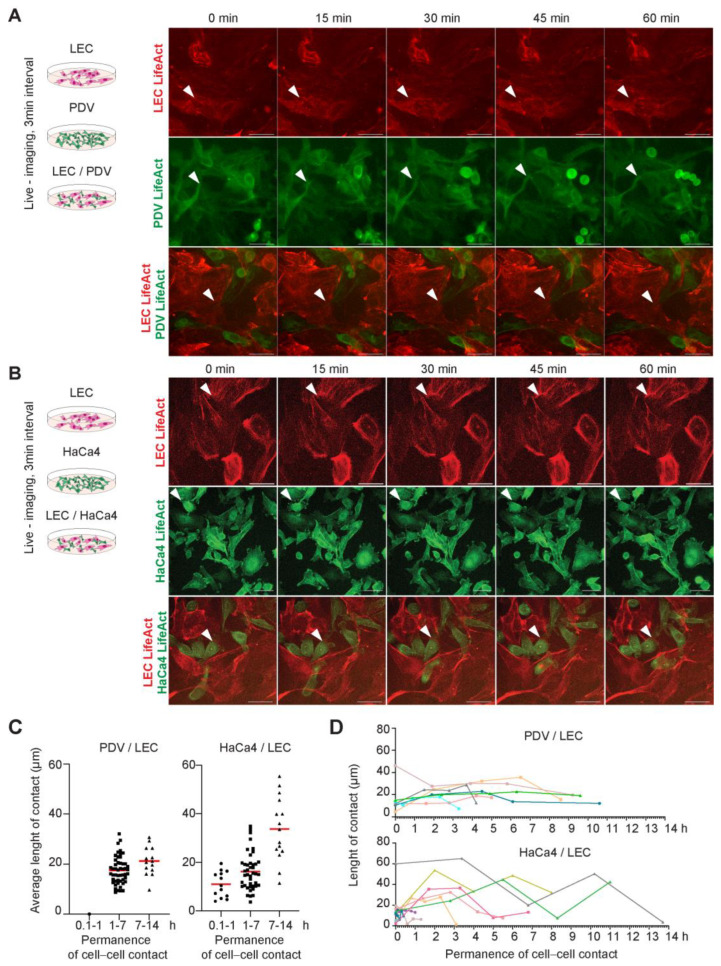
LEC and cancer cells interact dynamically over time, establishing heterotypic interactions. (**A**) Schematic illustrating the experimental strategy of PDV–initiated cancer cells and LECs live–imaging and representative stills from time–lapse imaging experiments showing homotypic and heterotypic cell–cell contacts over time (0–60 min) between PDV–LifeAct–GFP and LEC–LifeAct-RFP. Scale bar = 50 µm. White arrows. Representative cell–cell contact areas. (**B**) Schematic illustrating the experimental strategy of HaCa4–promoted cancer cells and LEC live–imaging and representative stills from time–lapse imaging experiments showing homotypic and heterotypic cell–cell contacts over time (0–60 min) between HaCa4–LifeAct–GFP and LEC– LifeAct–RFP. Scale bar= 50 µm. White arrows. Representative cell–cell contact areas. (**C**) Number of PDV and HaCa4 cells establishing cell contacts with LECs over 0.1–1 h, 1–7 h, and 7–14 h intervals and the average length of cell contacts (µm). (**D**) Graphs showing the surface length (µm) of cell–cell contacts at two or four different time points and the permanence of contacts over time (h).

**Figure 7 ijms-24-13615-f007:**
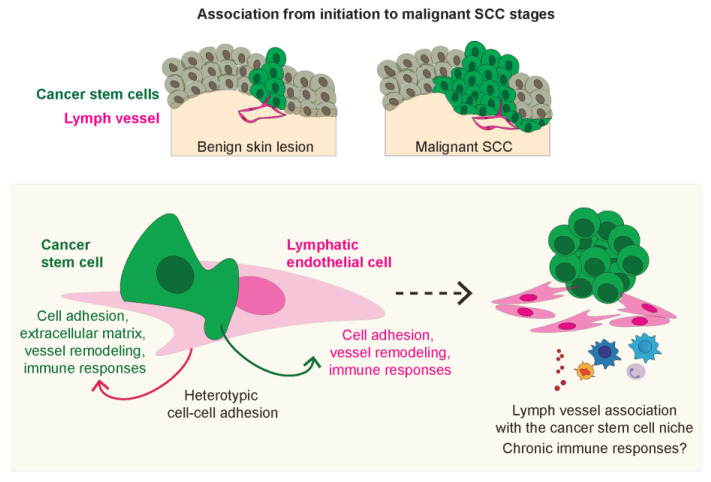
Lymph vessels associate with cancer stem cells from initiation to malignant stages of squamous cell carcinoma. Cancers stem cells and lymph endothelial cells establish heterotypic cell–cell adhesions that reciprocally influence gene expression programs related to cell adhesion, vascular regulation, and immunomodulatory responses independently of other cells in the complex tumor milieu. These interactions may foster tumorigenesis by sustaining signaling responses within the niche, potentially altering vessel permeability, and promoting chronic pro-tumorigenic immune responses.

## Data Availability

RNA sequencing data are accessible through GEO. Accession number: GSE227697. https://www.ncbi.nlm.nih.gov/geo/query/acc.cgi?acc=GSE227697. Accessed on 21 April 2023. Further information and requests for resources and reagents should be directed to the corresponding author.

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
