# Peer review of "Lymph Vessels Associate with Cancer Stem Cells from Initiation to Malignant Stages of Squamous Cell Carcinoma"

_ijms, 2023, doi:10.3390/ijms241713615_

Round 1
Reviewer 1 Report
The authors presented data demonstrating the importance of interaction between cancer stem cells and vessel endothelial cells in the initiating stage of skin squamous cell carcinoma development. The idea is novel, but the experimental designs may not be suitable for answering the questions raised by the authors. The main issues are listed as follows.
1. The authors used SOX2-expressing cancer cells in most in vitro experiments to present CSCs. However, these two cell lines (PDV and HaCa4) were cultured in 10% FBS-containing medium. How could they maintain CSCs in FBS-containing conditions? Can it represent CSCs by only SOX2 level?
2. How could the authors ensure that only CSCs interact with LEC if the authors did not use non-CSC cells as the control?
3. In the Methods of Bulk RNA-seq analysis, the authors described using HaCa4 cells. However, the data of PDV cells is also presented in Figure 3.
4. What were the differences of initiated PDV cells and the promoted HaCa4 cells in Figure 4(page 8, line 259)? The authors did not clearly define it.
5. The key factors for the interaction between CSCs and LECs were not directly identified by experiments, only being speculated by the RNA-Seq data. The authors should apply an antibody-blocking strategy for demonstrating the involvement of ITGA4 and VCAM1 in such interaction.
Reviewer 2 Report
This research analyzed the interaction of lymph vessels and skin squamous cell carcinoma cells. The relationship of these two types of cells did not resulted in an increase of the proliferation of carcinoma cells but changes in the gene expression, actin reorganization, and motility. The results are consistent with the idea that carcinoma cells use lymphatic vessels to escape the location and metastatize toward the lymph nodes. The authors may discuss in more details and provide differences between angiogenesis and lymph vessels proliferation. The manuscript is well written, it is easy to read, and to understand. The quality of the figures is high, and there are videos were the movement of carcinoma cells is seen.
Comments:
(1) Lines 78 to 85. Three types of skin lesions are described, benign papillomas, keratoacanthomas, and squamous cell carcinoma in mice. Nevertheless, in mice the lesions were induced using “chemical method”. Does the chemical method produce the 3 types of lesion? Is not papilloma usually more associated with human papilloma virus? How can the same chemical produce 3 different types of lesions?
(2) In Figure 1 text, could you please add in the description what K5 marker is identifying (keratinocytes)? Are K5+cells only located in the basel layer? In the immunofluorescence images the white color is only seen in the bottom. Conversely, red is seen in all cells, superficial and deep. Is SOX2 cytoplasmic or nuclear?
(3) In Figure 1, the SOX2+nuclear cells are located in the basal layers of the lesions inside the epithelial layer. Therefore, they are not related to hair follicle stem cells (HFSC), aren’t they?
(4) Does LYVE1 identify lymph vessels only, and not vascular vessels? Could you please epxlain why are lymph vessels more important in the pathogenesis than vascular vessels? In cancer, angiogenesis is very important, so I wonder why the authors target lymph vessels instead. Lymp vessels points out cell recirculation through the immune system.
(5) It is stated that LV distribute into the CSC niche. Are LV also present in that location in control samples?
(6) What are the properties of “K15-CrePR1; Wlsfl/fl (Wls∆K15)” mice? Do these mice lack SOX2+cells?
(7) Are LYVE1+LV increased in SCC than papilloma and keratoacanthoma?
(8) Line 129. “immortomouse”. What are the properties of this strain?
(9) In human samples, are LV increased compared to controls?
Round 2
Reviewer 1 Report
The authors have done the revisions according to most of the my questions. Only one minor revision is suggested.
1. Please provide some discussions about the importance and therapeutic innovations of the interaction between LECs and CSCs based on ITGA4 and VCAM1.
Author Response
We would like to express our gratitude to Reviewer 1 for their insightful feedback and careful consideration of our revisions. We acknowledge the importance of further discussing the significance and therapeutic implications of the interaction between lymphatic endothelial cells (LECs) and cancer stem cells (CSCs) mediated by ITGA4 and VCAM1, as pointed out by the Reviewer. In response to this suggestion, we have emphasized the potential therapeutic innovations that targeting this interaction presents, such as disrupting CSC maintenance and survival, immune evasion, and modulation of the microenvironment. See lines 394-402. We believe that this expanded discussion provides a deeper insight into the implications of the ITGA4-VCAM1 interaction and its relevance in developing innovative therapeutic strategies against CSC-mediated tumor progression.